# The lived and living experiences of having chronic pain and mental illness among Canadian veterans: A qualitative descriptive study

Umair Majid[1]*, Tom Hoppe[1], Nicholas Held[2,3], David Pedlar[2], Kerry Kuluski[1,4]

**1** Institute of Health Policy, Management and Evaluation, University of Toronto, Toronto, Canada, **2** Canadian Institute for Military and Veteran Health Research, Queen's University, Kingston, Canada, **3** Faculty of Health Sciences, McMaster University, Hamilton, Canada, **4** Institute for Better Health, Trillium Health Partners, Mississauga, Canada

* majidua@mcmaster.ca

## Abstract

Military Veterans are disproportionately affected by both chronic pain and mental illness, yet these conditions are often treated and understood in isolation. Existing clinical and policy frameworks rarely reflect the lived and living realities of Veterans who experience these forms of suffering as deeply interconnected. Understanding how Veterans themselves make sense of the relationship between chronic pain and mental illness is critical for developing more responsive, integrated care. This study asked Veterans how they understood the connection between their chronic pain and mental illness, and how this has enduring impacts on their everyday lives as Veterans in a civilian world. The objective of this study was to explore how Canadian Veterans perceive and navigate the connections between their chronic pain and mental illness in everyday life. Drawing upon interviews with 20 Canadian Veterans, they described chronic pain and mental illness as cyclical, dynamic, and shaped by personal and contextual factors. Chronic pain was seen to intensify mental illness, while mental illness amplified the experience of pain, producing a self-reinforcing loop of suffering. This cycle led to social withdrawal, loss of identity, diminished coping capacity, and reluctance to seek help. Many Veterans linked their pain to unresolved psychological trauma, interpreting it as a bodily expression of their military past. Conversely, a mental health diagnosis sometimes offered explanatory value and validation for their pain. Veterans described a constant negotiation between these conditions, mediated by the challenges of civilian reintegration and shifts in identity. Findings are discussed in relation to existing literature and theoretical frameworks to deepen our understanding of the entangled nature of chronic pain and mental illness in Veteran's lives.

**Data availability statement:** All relevant data are within the paper and its Supporting information files.

**Funding:** The study was funded by the Chronic Pain Centre of Excellence for Canadian Veterans (UM/KK). The funders had no role in study design, data collection and analysis, decision to publish, or preparation of the manuscript.

**Competing interests:** The authors have declared that no competing interests exist.

## Introduction

Chronic pain and mental illness have been increasingly linked through overlapping biopsychosocial mechanisms that place individuals at heightened risk for both conditions [1]. Veterans are a unique community that often face a distinct set of physical and psychological challenges upon leaving military service, placing them at higher risk for both chronic pain and mental illness compared to the general population. For instance, in a recent study of United States military Veterans, researchers found that a quarter of Veterans who had chronic pain had post-traumatic stress disorder (PTSD) and over half of those who had PTSD also had chronic pain [2]. Military training and operations can inflict injuries that lead to persistent pain, while the demanding and often traumatic nature of service can give rise to conditions such as PTSD, anxiety, and depression. Although many Veterans find relief and meaning in civilian life, for others, the intertwined burdens of physical and mental health difficulties continue to shape their everyday experiences long after discharge [3].

The comorbidity of chronic pain and mental illness holds particular significance for Veterans, as these two conditions can perpetuate and even magnify one another. Existing scholarship recognizes that chronic pain may exacerbate the symptoms of mental illness, just as mental illness can intensify pain sensations [4]. The Shared Vulnerability Model [5] posits that anxiety sensitivity — a predisposition to interpret bodily sensations as threatening — may serve as a common risk factor for both chronic pain and mental illness. Further research has highlighted the potential role of central sensitization, wherein neurological pathways involved in pain signaling become hypersensitive, thus amplifying pain perception and heightening vulnerability to PTSD symptoms [6,7]. In parallel, the Mutual Maintenance Model [8] explains how biases toward threatening or painful stimuli, fear-based avoidance behaviors, and recurring reminders of trauma can create a cycle that intensifies and sustains both chronic pain and PTSD. From heightened hypervigilance to the adoption of maladaptive coping strategies, both models underscore the intricate interplay of physiological, emotional, and cognitive factors that bind chronic pain to mental health conditions. While the Shared Vulnerability Model emphasizes pre-existing, shared etiological mechanisms that increase Veterans' risk for developing both chronic pain and mental illness, the Mutual Maintenance Model highlights the ongoing, reciprocal processes through which these conditions might interact and reinforce one another once established. This distinction is particularly salient for Veterans whose trauma exposure and injury may both initiate comorbidity and perpetuate symptoms through daily lived and living experiences.

Yet the lived and living experiences of these overlapping conditions are not fully understood nor investigated; in many cases, research has examined these issues in isolation, focusing either on chronic pain or on mental health symptoms. However, when both chronic pain and mental illness are present, Veterans can be drawn into cycles of limited mobility, social isolation, emotional distress, and diminished quality of life, all of which demand a holistic lens that captures how these conditions converge in everyday routines, relationships, and environments [3]. Much of the existing literature has relied on quantitative or symptom-focused approaches, which, while

informative, provide limited insight into how Veterans interpret, manage, and live with this comorbidity on a day-to-day basis [9–11]. Relatively little qualitative research has explored how Canadian Veterans understand the interconnected nature of chronic pain and mental illness or how these conditions shape their post-service lives within civilian life. Articulating Veterans' own perspectives addresses a critical gap in the literature by extending theoretical models of comorbidity into lived/living experience and informing more responsive, Veteran-centred care.

This study asked Veterans how they understood the connection between their chronic pain and mental illness, and how this has enduring impacts on their everyday lives as Veterans in a civilian world. The objective of this study was to explore how Canadian Veterans perceive and navigate the connections between their chronic pain and mental illness in everyday life. The central research questions guiding this study were: how do Canadian Veterans perceive and navigate the relationship between their chronic pain and mental illness? In what ways do these interconnected conditions shape their lived and living experiences in civilian life?

### Military identity, mindset, and culture

In exploring how Canadian Veterans articulate the connection between chronic pain and mental health, it is crucial to differentiate between the concepts of *identity*, *mindset*, and *culture*, which are often invoked interchangeably but operate at distinct levels of human experience.

1. **Identity** refers to the multifaceted and evolving sense of self, encompassing how individuals perceive themselves and how they are socially recognized, particularly in relation to roles, values, and histories [12,13]. It is both internal and relational and often reshaped in response to life transitions such as military discharge or the onset of chronic pain [14].

2. **Mindset** is a cognitive-affective orientation — such as optimism, stoicism, or resilience — that influences how individuals interpret and respond to challenges [15,16]. While it may inform coping behaviors and pain management strategies [17], mindset does not define the broader narrative of selfhood.

3. **Culture** refers to the shared norms, values, and practices of a group or society; it provides the symbolic and institutional context within which both identity and mindset are formed and expressed [18,19]. For Veterans, cultural norms around masculinity, duty, and endurance often intersect with personal mindsets and identity transformations, shaping how they interpret and communicate their pain and psychological distress.

## Methods

### Ethics statement

The University of Toronto Research Ethics Board approved this study. All participants provided written consent via email or verbal consent on recording before the interview started and the University of Toronto Research Ethics Board approved this procedure.

### Approach

This study employed a qualitative description approach [20] to examine how Canadian Veterans experience the interplay of chronic pain and mental illness in their day-to-day lives. While the interviews were conducted as part of a broader narrative inquiry study focusing on identity in Canadian Veterans with chronic pain, the analysis presented here draws specifically on the portions of those interviews in which Veterans were asked targeted questions about the connection between their mental illness and chronic pain and how having both impacts their everyday lives as Veterans in a civilian world. A qualitative description approach was deemed most suitable for capturing a rich yet pragmatic account of Veterans' experiences without imposing an overly interpretive or theoretical framework that may limit a deep appreciation of the lived and living realities of comorbidities.

## Sample and recruitment

All Veterans who participated in the broader narrative inquiry study answered questions related to the intersection between chronic pain and mental illness that informed this paper. In the broader narrative inquiry study, eligible participants were Canadian Veterans aged 18 or older with chronic pain who had previously agreed to share their experiences of military and civilian life as part of the project. Veterans currently serving in the military and non-Canadian forces were excluded. Recruitment sought a convenience sample of Veterans facilitated through the Chronic Pain Centre of Excellence for Canadian Veterans listserv. An email was sent out to the listserv and interested Veterans reached out directly to the research team to learn more about the study and how they could participate. The sample size (n = 20) was selected to capture a range of complex and diverse experiences for the broader narrative inquiry study rather than to achieve thematic saturation. This approach was appropriate for the exploratory nature of this study, which aimed to explore variability in how Veterans understood and lived with both chronic pain and mental illness, rather than to produce an exhaustive or homogeneous account of these experiences.

## Data collection

Data were gathered between August and December 2024 using in-depth, semi-structured interviews. Participants were asked about multiple facets of their military service and transition to civilian life, including how chronic pain affected their daily routines, social relationships, and personal well-being. As part of these initial interviews, the researchers asked questions specifically about how participants perceived the relationship between their chronic pain and mental illness. Interviews lasted approximately 60–90 minutes and were conducted primarily by the lead researcher, with a bilingual co-researcher conducting interviews when participants preferred to speak French. Immediately following each interview, the interviewers wrote reflexive memos to capture initial insights, questions, and possible thematic directions for subsequent analysis.

Given that discussions of chronic pain and mental health evoked distressing memories or emotional discomfort, several ethical safeguards were embedded throughout the data collection process. Before participation, Veterans were informed of the sensitive nature of the interview topics and reminded that participation was voluntary, that they could decline to answer any question, and that they could pause or withdraw from the interview at any time without consequence. Interviewers were attentive to verbal and non-verbal cues of distress and were prepared to slow the pace, redirect the conversation, take breaks, or stop the interview if needed. When participants expressed emotional discomfort, interviewers responded with empathy and grounding techniques using a trauma-informed interviewing protocol developed by the research team and Veteran partner, and participants were offered breaks or the option to reschedule. All participants were provided with information about mental health and Veteran-specific support resources at the conclusion of the interview, including instructions on how to access additional support if distress emerged after participation. These measures were implemented to minimize potential risks and to support participants' well-being throughout the research process.

## Data analysis

All interviews were audio-recorded and transcribed verbatim. A thematic analysis approach [21,22] consistent with qualitative description was used to organize, interpret, and represent the data. First, two members of the research team (one being a Veteran partner) independently read four transcripts in their entirety to gain familiarity with the content. They then identified preliminary codes relevant to the intersection of chronic pain and mental illness, discussing them in team meetings to resolve discrepancies and refine the coding structure. Through an iterative process, the codes were grouped into broader themes that captured key patterns in how Veterans perceived and navigated the challenges of simultaneous chronic pain and mental illness. Regular analytic discussions ensured that emerging themes remained grounded in participants' accounts. This approach allowed the researchers to produce an integrated portrayal of Veterans' experiences, highlighting how chronic pain and mental illness interact to shape their everyday lives.

### Veteran partner reflexivity

The inclusion of a Veteran partner on the research team brought experiential knowledge of military service, chronic pain, and mental health that informed the interpretation of participants' accounts. This perspective supported analytic sensitivity to the nuances of Veterans' language, contextual meanings, and everyday realities that may not be readily apparent to non-Veteran researchers. At the same time, the Veteran partner engaged in ongoing reflexive dialogue with the broader research team to critically examine how shared experiences could shape interpretations.

## Findings

Twenty Canadian Veterans with concurrent diagnoses of chronic pain and mental illness were interviewed for this study. In terms of chronic pain, all Veterans reported multiple locations for their chronic pain, but primarily their back (n = 19, 95%), knees or legs (13, 65%), ankles or feet (n = 7, 35%), and hips (n = 6, 30%). For mental illness, all Veterans had a mental illness diagnosis, with PTSD (n = 13, 65%) being the most common, followed by major depressive disorder (n = 8, 40%) and anxiety disorders (n = 4, 20%). A complete list of demographic characteristics is shown in Table 1. Synthesized findings are available in S1 Data.

We asked each Veteran to describe the relationship between their chronic pain and mental illness and how having both impacts their lives. One veteran (n = 1) believed there was no relationship between chronic pain and mental health, and the remaining 19 Veterans believed there was a relationship, though they explained the relationship and its impact on their everyday life to different degrees. Seven Veterans believed that their mental illness intensified their chronic pain, seven Veterans said their chronic pain adversely affected their mental health and well-being, and four said both chronic pain and mental illness exacerbate each other in a bidirectional relationship. It is important to note that several Veterans provided overlapping accounts that speak to more than one kind of relationship between chronic pain and mental illness. The sections below describe how Veterans explain the relationship between mental illness and chronic pain and how having both impacts their daily lives as Veterans in a civilian world. A summary of themes and subthemes is shown in Table 2.

### How mental health and illness affect chronic pain

For Veterans in this study, mental illness actively shaped and intensified their chronic pain. Veterans recounted that conditions such as PTSD, anxiety, and depressive disorders emerged before or concurrently with the onset of their chronic pain; for example, one Veteran received a diagnosis of major depressive disorder before realizing he had chronic pain, which made him suggest during the interview that unresolved psychological distress predisposed him to chronic pain. Central to Veterans was a belief that mental illness manifests physically; as one Veteran described chronic pain as a "manifestation of mental health challenges," arguing that physical pain should be acknowledged and treated rather than dismissed because the cause is mental illness.

**Mental illness diagnosis, compensation, and treatment provided an explanation.** Getting a mental illness diagnosis gave some Veterans confidence and clarity; a mental illness diagnosis offered clarity and an explanation of how they felt, and this gave them the confidence to receive care for their mental illness, chronic pain and other conditions.

> "Yeah, [depressive disorder] didn't really impact my pain level at all, but it certainly explained why I think the way that I think a lot of it, and gave me some confidence that there was help there available, and they were going to finally look after me properly" (7149).

Another Veteran was suggested by a representative of a government agency to request compensation for mental illness alongside their existing chronic pain, suggesting that both health issues were related and physical injuries often come from underlying mental health challenges.

**Table 1. Demographic and social identity characteristics of veterans.**

| Characteristic | | n (%) |
| --- | --- | --- |
| Gender | Man | 15 (75.0) |
| | Woman | 5 (25.0) |
| Age | Mean | 57.16 years |
| | Median | 56 years |
| | 40-49 | 3 (15.0) |
| | 50-59 | 9 (45.0) |
| | 60+ | 8 (40.0) |
| Race/Ethnicity | White | 17 (85.0) |
| | Non-White | 3 (15.0) |
| | French | 2 (10.0) |
| Province | Alberta | 6 (30.0) |
| | British Columbia | 2 (10.0) |
| | Nova Scotia | 2 (10.0) |
| | Ontario | 6 (30.0) |
| | Quebec | 3 (15.0) |
| | Saskatchewan | 1 (5.0) |
| Rurality | Urban | 9 (45.0) |
| | Suburban | 4 (20.0) |
| | Rural | 7 (35.0) |
| Highest Education | High school | 6 (30.0) |
| | College | 6 (30.0) |
| | University | 5 (25.0) |
| | Graduate | 3 (15.0) |
| Years of Service | Mean | 24.08 |
| | Median | 24 |
| | 0-9 | 2 (10.0) |
| | 10-19 | 4 (20.0) |
| | 20-29 | 7 (35.0) |
| | 30+ | 7 (35.0) |
| Year Released | Median | 2016 |
| | Mode | 2019 |
| | ≤1999 | 3 (15.0) |
| | 2000-2009 | 3 (15.0) |
| | 2010-2019 | 12 (60.0) |
| | ≥2020 | 2 (10.0) |
| Military Units | Army | 15 (75.0) |
| | Air Force | 5 (25.0) |
| | Navy | 3 (15.0) |
| Military Rank | Corporal | 5 (25.0) |
| | Sargent | 7 (25.0) |
| | Master Warrant Officer | 1 (5.0) |
| | Captain | 1 (5.0) |
| | Major | 5 (25.0) |
| | Colonel | 1 (5.0) |

*(Continued)*

**Table 1.** (Continued)

| Characteristic | | n (%) |
|---|---|---|
| Chronic Pain Location | Back | 19 (95.0) |
| | Knees/Legs | 13 (65.0) |
| | Ankles/feet | 7 (35.0) |
| | Hips | 6 (30.0) |
| | Neck | 5 (25.0) |
| | Elbows/Arm | 5 (25.0) |
| | Wrist/Fingers | 3 (15.0) |
| | Shoulder | 2 (10.0) |
| | Jaw | 1 (5.0) |
| Mental Illness Diagnoses | PTSD | 13 (65.0) |
| | Depressive Disorders | 8 (40.0) |
| | Anxiety Disorders | 4 (20.0) |
| | Insomnia | 3 (15.0) |
| | OCD | 2 (10.0) |
| | Migraines | 2 (10.0) |
| | Conflict Morality | 1 (5.0) |

*Migraines* were reported by participants as a chronic pain condition and were included when Veterans described them as persistent, disabling, and co-occurring with mental health symptoms.

*Moral injury* (referred to as conflict morality by one participant) was included as a mental health–related condition when Veterans identified it as a source of enduring psychological distress linked to their military experiences.

"I did talk to someone from [organization], and they, like, suggested, oh, well, maybe you should have a claim in for some, like, psychological issues. Maybe you should go see a counselor. And then I thought, oh, really, you think I should kind of…Is there something wrong with my mind? Well, I wasn't, like, upset about it or anything. It was just more of a oh, enlightenment, like, oh, maybe I should like…thank you [organization] for suggesting it that they would be linked together based on the information that I had wrote to support [for chronic pain]. So, I don't know if they read it or link those two together, or if they just know that [chronic pain is sometimes also mental health related injuries that aggravate them, so maybe the mental health injury should have came first and then the other physical injuries" (6866).

To emphasize the effect of mental illness on chronic pain, a Veteran recounted the transformative impact of ketamine treatment for PTSD on his chronic pain and said:

**Table 2.** Summary of themes and subthemes.

| Theme | Sub-Themes |
|---|---|
| How Mental Health and Illness Affect Chronic Pain | • Mental illness diagnosis, compensation and treatment provided an explanation<br>• Everyday stress exacerbates chronic pain |
| How Chronic Pain Affects Mental Illness and Health | • Chronic pain drains energy to seek care and engage in activities<br>• Chronic pain can provide accidental clarity about mental health |

"my brain started to function a lot better [after ketamine treatment]. I mean, I'm still not, 'normal,' but it's way better than where I was, and the pain came down with it…So to me, they're definitely connected. They started at the same time, they reduced at the same time" (7864).

This account emphasizes the temporal synchrony between mental health improvement and pain relief and raises important points about the intersection and synergies of physical and psychological treatments for Veterans.

**Everyday stress exacerbates chronic pain.** Veterans described how stress — whether from crowded environments like stores or the enduring presence of a military mindset at work — precipitates or intensifies chronic pain symptoms. One Veteran vividly described how the stress of navigating busy public spaces triggered debilitating back pain:

"I still had that stress, like, I have to be there, I can't be late, I have to do this, you know, all that military mindset, the 'I need to perform' mentality. It has to show results. All of that just increased my anxiety. I had more back pain, more ankle pain, headaches. The days I went there, I'd get headaches. The days I had to go there, I'd just think, 'Wow, this is crazy. It's crazy how connected everything is'" (9443).

Another Veteran highlighted how the same "I need to perform" mentality in high-pressure work environments led to exacerbated anxiety, back pain, and headaches. Another Veteran explained how anxiety and lethargy that was associated with having a mental illness indirectly accentuated the physical sensations of pain, especially when routine activities were more challenging to do. When this Veteran was asked whether mental illness caused chronic pain or vice versa, they said:

"I would say that it [pain] went up because I like when I get anxious, or any of that, I like, I hold that energy in and it manifests itself in me not looking after myself" (1652).

Another Veteran went further to describe why they think stress exacerbates pain by explaining how everyday job-related stress releases cortisol that gets "trapped" in muscles and joints, which then causes and exacerbates chronic pain.

"I guess stress releases a lot of cortisol. Cortisol gets trapped in your muscles and joints, and that's what causes a lot of your pain. Totally makes sense. I'm kind of a logical person, so it does make sense" (7864).

At the same time, this Veteran expressed skepticism toward the idea that pain is solely a byproduct of psychological processes: "because this theory is being pushed more, we're being ignored, again, marginalized, because [they say my pain] is just in my head" (7864). They were concerned about healthcare providers inadvertently stigmatizing physical chronic pain because of a belief that it was caused by underlying mental illness.

Another Veteran reflected on external conditions that could have, with mental illness, led to increased chronic pain, such as being alone and having to move:

"but what I thought would be normal of just missing my family because, you know, they're in [city], and I'm in [city], that I was more depressed. I was not energetic. I was very lethargic. I, you know, all of that kind of stuff. So when you're not moving, you know, it you're all of your injuries just become accentuated" (1652).

### How chronic pain affects mental illness and health

Compared to those who described how mental illness affected their chronic pain, an equal number of Veterans indicated that their chronic pain exerts a profound toll on their mental health by reducing their energy levels, diminishing their sense of self, and isolating them from previously valued activities.

**Chronic pain drains energy to seek care and engage in activities.** For Veterans in this study, chronic pain was not understood as an isolated physical condition coming from military injuries but an ongoing experience that drains emotional reserves and disrupts daily life. One veteran (3528) observed that while his mental resilience allowed him to address physical health challenges, the persistent nature of his pain ultimately limited his capacity to engage with his mental health needs. His chronic pain not only affected his confidence but also instilled a persistent fear that was rooted in an awareness of his physical limitations being in stark contrast to the stoic image of himself developed during military service and held since then.

"My mental health never stopped me from fixing my physical health, but my physical health was a huge factor in how much success I could have in dealing with my mental health…the physical pain affected my mental health. Like affected how confident you feel, how fearful you are. Like, there was a long time I never felt safe. I didn't feel safe about how I was perceived. I didn't feel safe about if something bad were to happen, would I have the strength and physical capability to deal with that situation?" (3528).

Several Veterans underscored how chronic pain can lead to a withdrawal from activities that once provided meaning and pleasure. For instance, Veterans described the frustration of beginning a walk around their house only to be forced to turn back after a few minutes due to overwhelming pain or even the demoralizing experience of being unable to complete everyday tasks such as grocery shopping or household chores because of their chronic pain.

One vivid account from a Veteran showed the spiral effect that comes from chronic pain. This Veteran explained how chronic pain saps energy and enjoyment from life, which in turn deepens feelings of depression and isolation, exacerbating not only their mental illness but also their chronic pain. This spiral effect was poignantly captured by a Veteran who recounted how chronic pain "drains your energy" and "saps your ability to do things," thus feeding into an emotional cycle of despair and social withdrawal.

"Chronic pain constantly wears on you, and I think that's another thing that people don't understand, is how badly it just saps your energy because you're fighting every day. It's constant. So that takes its toll emotionally on you, because it not only impacts your physical well being, but your emotional well being, of things that you used to be able to do you can't do now...because that chronic pain drains your energy, it drains your ability to do things, and then that just feeds on top of the depression, the PTSD, [and] relationships start to suffer. Inside your head spirals, and that's what gets hard to get out of. And then you start isolating yourself more, because, well, every time I go out, people bitch I'm too loud. People bitch on this and thats fine, I just won't freaking go. And then pretty soon that whole social circle, or your social environment, gets smaller, smaller, smaller, and pretty soon it's just you and your house" (4119).

The constant preoccupation with physical pain frequently left Veterans with little mental energy to pursue social or therapeutic activities. The pervasive impact of chronic pain often resulted in a curtailed ability to participate in activities that once fostered social interaction and personal fulfillment. One Veteran, for example, described the crushing reality of having to "fix" his pain every morning before he could even begin to address his PTSD symptoms. While sometimes necessary for survival, this compartmentalization further entrenched isolation and reinforced a sense of loss of not only physical capacity but also of identity, purpose and mindset.

The impact of chronic pain on sleep and daily routines further compounded mental health challenges. One Veteran explained that the severe, unyielding nature of his pain not only disrupted his sleep patterns but also precipitated a cascade of negative emotions (much like the Veteran above), leading him to describe the interplay as "just a big circle" where mental health and physical pain intensify one another.

"It just seemed like the chronic pain compounded my mental health. My mental health didn't help my chronic pain, and it was just a big circle" (7139).

"You have all these experiences [from the military], and I put them in a packsack and carry on. I went through so much, both from the military and Red Cross, but from the military, you put them away and you carry them. You let them out [on] Remembrance Day. I let them out at different times. You think about them. You pack them up, put them back in your pack and zip it up and carry on. I just found as the chronic pain got worse, it's almost like the zipper broke and things would fall out when I didn't want it to fog, and then suddenly I'd be in traffic, or suddenly I'd be with my partner, and the sights and sounds or the smells would just trigger me" (7139).

**Chronic pain can provide accidental clarity about mental health.** Much like Veterans from the previous section who said that a mental illness diagnosis offered clarity and an explanation, Veterans in this study also discussed how having chronic pain led to a mental illness diagnosis, sometimes accidentally. Chronic pain treatment caused one Veteran to have suicidal thoughts that led to a diagnosis of PTSD and another to have sleep issues that also eventually led to a PTSD diagnosis. Both Veterans linked this relationship to the disjointedness of mental health and physical treatment they received in the military and civilian life.

"So that disjointedness, that lack follow up, had been hiding mental problems, and once the doctor, my psychologist, explained to me that I had PTSD, all of the things, all of the pennies dropped into place, the lights went on, I said, now I know, and now I understand. And my life became clear" (2791).

## Discussion

This study showed that chronic pain and mental illness are rarely experienced in isolation by Canadian Veterans. Rather, they are entwined in deeply personal, cyclical, and contextually influenced ways. When asked how they understood each of their conditions, Veterans often understood one condition through the lens of the other. A common finding across Veterans was the circularity of suffering: chronic pain worsened mental illness, which in turn magnified the experience of pain and led to social withdrawal, loss of identity, and reduced help-seeking. Veterans described a constant negotiation between their chronic pain and mental illness, where the emergence or worsening of one directly impacted the other. For many Veterans, having a mental illness heightened their awareness of bodily discomfort, made pain more difficult to cope with, or even gave rise to new or intensified pain symptoms. Conversely, the unrelenting and debilitating nature of chronic pain depleted emotional reserves, restricted activity, and diminished social engagement, which further eroded mental health. At the same time, a mental illness diagnosis brought explanatory power and, for some, validation for their pain; for example, some Veterans traced their pain back to unresolved psychological trauma, seeing it as an echo of earlier military experiences. In this section, we explore extant literature, including models and frameworks, and how they can deepen our understanding of the impact of having both chronic pain and mental illness on the lives of Veterans navigating military mindset and identity change.

### Understanding comorbidities through theoretical and intersectional lenses

Veterans' experiences reported in this study echo and extend longstanding models that seek to explain the interconnected nature of chronic pain and mental illness. Military culture, mindset, and identity create the conditions that can contribute to chronic pain and mental illness. The **Shared Vulnerability Model** [5] suggests that individuals with heightened anxiety sensitivity or emotional dysregulation are more likely to develop both chronic pain and PTSD, due to a common predisposition toward amplifying internal sensations and threat perceptions. This theoretical framing was reflected in Veterans' accounts that located the origins of their pain in psychological distress or trauma from the military that had left lasting

impacts on the body. Several Veterans described pain emerging after episodes of depression and anxiety. However, these Veterans did not experience this as merely a matter of shared sensitivity; they saw it as a cumulative process of embodiment, where years of internalized stress and emotion eventually settled in the body, leading to chronic pain and/or mental illness. Therefore, Veterans' experiences suggest that psychological predispositions are compounded by social, cultural, and institutional forces that shape how suffering becomes physically inscribed.

More specifically, some Veterans in this study explained how their chronic pain contributed to their mental illness, and the **Allostatic Load Model** [23] suggests that this can happen through the cumulative burden of chronic pain, which can lead to physiological dysregulation that then weakens the body's capacity to adapt and predisposes individuals to mental illness. Veterans' experiences echo this model, particularly in how day-to-day stress related to performance expectations, civilian work environments, or internalized military values, can exacerbate both chronic pain and mental illness symptoms. Veterans experienced stress not as an episodic state, but rather as a persistent one carried over from military to civilian life. Veterans described being "wound tight" or having pain flare during stressful moments, and one Veteran explained how stress-induced cortisol became "trapped" in joints and muscles. These embodied understandings align with the model's core principle that chronic pain has biological consequences, but they also highlight a practical gap: care systems often fail to address chronic pain as a treatment priority. Veterans' experiences in this study highlight the need for care systems that acknowledge the impact of service, transition, and social disconnection, rather than just the symptoms that emerge downstream.

When both chronic pain and mental illness are present, the **Mutual Maintenance Model** [8] suggests that mental illnesses like PTSD and chronic pain reinforce one another through overlapping psychological and behavioral processes such as hyperarousal, attentional bias toward threat and pain, avoidance of activity, and catastrophic thinking. Veterans in this study gave rich accounts that align with this model, particularly in how chronic pain and mental illness interacted to create a mutually reinforcing feedback loop that eroded emotional stability and functional capacity. Several Veterans described how the stress of anticipating chronic pain reactivated military trauma they experienced, while others noted how psychological distress — especially in crowded or overstimulating environments — intensified their chronic pain.

Having chronic pain and mental illness has the potential to disrupt identity more so than if one were only present. The **Embodied Disruption Framework** [24] suggests that chronic pain and mental illness can work together to fracture a person's sense of bodily coherence, identity, and continuity with their past selves. This concept was powerfully illustrated in Veterans' accounts of how both chronic pain and mental illness disrupted not only their functioning but also their fundamental self-concept. Many saw themselves as physically diminished versions of their former military selves, no longer able to live up to the disciplined, high-performing identity that had once been central to their sense of purpose. The transition to civilian life, already a complex and disorienting process, was further complicated by the dual burden of chronic pain and mental illness. As a result of this dual burden, Veterans described feeling "unsafe" in their bodies, incapable of protecting themselves or participating in the civilian world. One Veteran noted that his social world had contracted until it was him alone in his house, underscoring how chronic pain and mental illness together dislocated him from community, purpose, and belonging. Conflicting values and beliefs between military and civilian lives increase stress and impact mental health. Healing from such consequences requires more than physical or emotional restoration; it demands a reconstitution of meaning, identity, and connection that considers the interdependent impacts of chronic pain and mental illness.

What is critical from applying these models and frameworks to our study findings is not only to view these conditions through biomedical or psychological models, but to explore and potentially interrogate the *intersectionality* of how they are experienced and embodied. Veterans' stories were shaped by more than diagnoses. Their experiences were profoundly contextual, shaped by who they were, what they had endured, and their social and economic circumstances. The deeply ingrained "need to perform" mentality, as a case in point, continues to influence Veteran experiences of both stress and self-worth in civilian workspaces that perpetuate cycles of physical and emotional strain.

There is a pressing need for more research that takes seriously the intersecting nature of comorbid conditions — not simply as clinical co-occurrences, but as lived and living entanglements. Such work must explore how one condition is understood through the prism of another by the person affected, how treatments for one condition impact the trajectory of the other for the person affected, and how these interdependencies affect the lives not only of patients but also their families and support networks. An intersectional perspective that attends to the socio-cultural, institutional, and historical forces shaping Veterans' lives can help avoid reductionist framings and ensure that interventions speak to the full scope of their realities. We have tried to address these points by exploring the intersection of chronic pain and mental illness under the purview of how military identity changes from military service to civilian life.

## Implications

The findings of this study have important implications for Veteran health systems, particularly the need to move beyond siloed, diagnosis-driven models of care toward approaches that recognize chronic pain and mental illness as interdependent and mutually shaping conditions. Veterans' accounts show that treating pain and mental health in isolation risks overlooking the cyclical, embodied ways these conditions interact to erode identity, functioning, and social connection. Health systems serving Veterans must therefore adopt integrated, trauma-informed models of care that explicitly acknowledge how military service, identity, and transition to civilian life shape both pain and psychological distress. This includes care pathways that enable mental health and pain services to work collaboratively and support the restoration of meaning, agency, and belonging. Veterans' narratives further suggest that clinical encounters should create space for Veterans to articulate how they understand the origins and interconnections of their pain and mental illness, rather than imposing narrow biomedical explanations that may feel incomplete or invalidating. Incorporating these lived and living perspectives into assessment, care planning, and rehabilitation may help reduce disengagement from services and improve continuity of care. Ultimately, Veteran health systems that foreground the relational, contextual, and identity-based dimensions of comorbidity may be better positioned to support healing that extends beyond symptom reduction to encompass social reintegration and long-term well-being.

## Limitations of this study

Several limitations should be considered when interpreting the findings of this study. First, this analysis was not intended to provide an in-depth or interpretive examination of Veterans' lived and living experiences. Rather, a qualitative descriptive approach was intentionally employed to address a focused research objective nested within a broader narrative inquiry. While this approach allowed for a pragmatic and accessible account of how Veterans understand and navigate the intersection of chronic pain and mental illness, it necessarily limits the depth of interpretive insight that could be generated. As such, this study should be viewed as an important first step that establishes the relevance of this comorbidity in Veterans' everyday lives and points toward the need for future research using more deeply interpretive methodologies, such as phenomenology, to further explore how chronic pain and mental illness shape Veterans' lived and living experiences over time.

Second, the sample was predominantly composed of White male Veterans. Experiences of comorbidity are likely shaped by intersecting forms of marginalization, including gender, race, and access to social and health resources, all of which were underrepresented in this study. The findings may not fully capture how the intersection of chronic pain and mental illness is experienced by Veterans across diverse gendered, racialized, and social identities. Future research should prioritize more diverse and purposive sampling strategies to examine how chronic pain and mental illness are experienced within broader social and structural contexts among underrepresented Veteran populations.

## Supporting information

**S1 Data. Synthesized, de-identified data.**
(DOCX)

## Author contributions

**Conceptualization:** Umair Majid, Tom Hoppe, Nicholas Held, David Pedlar, Kerry Kuluski.

**Data curation:** Umair Majid, Tom Hoppe, Kerry Kuluski.

**Formal analysis:** Umair Majid, Tom Hoppe.

**Funding acquisition:** Umair Majid, Tom Hoppe, David Pedlar, Kerry Kuluski.

**Investigation:** Umair Majid, Nicholas Held, David Pedlar, Kerry Kuluski.

**Methodology:** Umair Majid, Nicholas Held, David Pedlar, Kerry Kuluski.

**Project administration:** Umair Majid, Kerry Kuluski.

**Resources:** Umair Majid, Kerry Kuluski.

**Software:** Umair Majid, Kerry Kuluski.

**Supervision:** Umair Majid, Kerry Kuluski.

**Validation:** Umair Majid, Tom Hoppe, Nicholas Held, David Pedlar.

**Visualization:** Umair Majid.

**Writing – original draft:** Umair Majid.

**Writing – review & editing:** Umair Majid, Tom Hoppe, Nicholas Held, David Pedlar, Kerry Kuluski.

## Acknowledgments

Kuluski is supported by the Dr. Mathias Gysler Research Chair in Patient and Family Centred Care funded by the Trillium Health Partners Foundation.

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
