## [Decision Letter · Decision Letter 0]

19 Dec 2025

PMEN-D-25-00428

The Lived and Living Experiences of Having Chronic Pain and Mental Illness Among Canadian Veterans: A Qualitative Descriptive Study

PLOS Mental Health

Dear Dr. Majid,

Thank you for submitting your manuscript to PLOS Mental Health. After careful consideration, we feel that it has merit but does not fully meet PLOS Mental Health’s publication criteria as it currently stands. Therefore, we invite you to submit a revised version of the manuscript that addresses the points raised during the review process.

We look forward to receiving your revised manuscript.

Kind regards,

Marc Eric Santos Reyes

Academic Editor

PLOS Mental Health

Journal Requirements:

1. In the ethics statement in the Methods, you have specified that verbal consent was obtained. Please provide additional details regarding how this consent was documented and witnessed, and state whether this was approved by the IRB.

Additional Editor Comments (if provided):

Reviewers' comments:

Reviewer's Responses to Questions

**Comments to the Author**

1. Does this manuscript meet PLOS Mental Health’s publication criteria ? Is the manuscript technically sound, and do the data support the conclusions? The manuscript must describe methodologically and ethically rigorous research with conclusions that are appropriately drawn based on the data presented.

Reviewer #1: Yes

Reviewer #2: No

Reviewer #3: Yes

2. Has the statistical analysis been performed appropriately and rigorously?

Reviewer #1: N/A

Reviewer #2: N/A

Reviewer #3: N/A

3. Have the authors made all data underlying the findings in their manuscript fully available (please refer to the Data Availability Statement at the start of the manuscript PDF file)?

Reviewer #1: Yes

Reviewer #2: No

Reviewer #3: Yes

4. Is the manuscript presented in an intelligible fashion and written in standard English?

Reviewer #1: Yes

Reviewer #2: Yes

Reviewer #3: Yes

Reviewer #1: This is an engaging and methodologically sound qualitative manuscript that provides an important contribution to understanding the intersection of chronic pain and mental illness among Canadian Veterans. The integration of participants’ narratives with theoretical models adds considerable depth and clinical relevance. The following suggestions are intended to strengthen an already strong manuscript.

Major Strengths

- The study captures rich, authentic narratives that humanize Veterans’ lived and living experiences.

- The theoretical framing is thoughtful and multidimensional, bridging biomedical, psychological, and sociocultural perspectives.

- The methodological rigor is evident, and the inclusion of a Veteran partner in the analysis process is a notable strength.

- The findings are highly relevant to integrated and trauma-informed care approaches for Veterans.

Suggestions for Revision

- Clarify Theoretical Integration: Explicitly differentiate how the Shared Vulnerability and Mutual Maintenance models each uniquely apply to Veterans’ experiences.

- Expand Intersectional Discussion: Consider acknowledging gendered and racialized aspects of the sample (85% White, 75% male) to situate findings within broader intersectional contexts.

- Reflexivity/positionality: Briefly elaborate on how the Veteran co-researcher’s perspective shaped data interpretation

- Findings Section: Condense slightly and consider using subhead summaries or a visual model to illustrate the cyclical pain–mental illness feedback loop.

- Discussion: Add a short paragraph on practical implications for Veteran health systems (e.g., integrated care, biopsychosocial rehabilitation, or interdisciplinary treatment).

- Editorial Polishing: Minor proofreading and clarification (e.g., consistency in tense and citation style) will further strengthen clarity.

Reviewer #2: The rationale for examining the lived and living experiences of military veterans requires further strengthening. While the topic is relevant, the arguments that build toward this focus remain underdeveloped. It is recommended that the authors elaborate on these points by clearly articulating why such experiences warrant focused scholarly attention and how they contribute to existing knowledge.

Relatedly, the research gap is not sufficiently articulated. The manuscript would benefit from a clearer and more explicit statement of the gap that this study seeks to address, particularly in relation to prior empirical or theoretical work on military veterans’ experiences.

The central research question is also not clearly evident. To improve coherence and direction, the authors are encouraged to explicitly state the main research question (and/or sub-questions) guiding the inquiry.

The study indicates the use of a qualitative descriptive approach; however, the emphasis on participants’ lived and living experiences appears to also be closely aligned with a phenomenological orientation. The authors may wish to reconsider their methodological framing and clarify whether a phenomenological approach should be incorporated or explicitly justified as not being used.

Further clarification is needed regarding participant selection. The inclusion and exclusion criteria should be clearly defined, and the manuscript should specify how participants were recruited or accessed. Providing this information will enhance the transparency and rigor of the study.

In addition, the demographic characteristics of the participants are not adequately presented in the manuscript but was only presented in a table as part of the appendices. A clearer presentation of relevant demographic information, such as age range, length of service, rank, or other pertinent characteristics, as they would help contextualize the findings.

It is also important to note that military veterans may be considered a vulnerable population, particularly given the sensitive nature of the topics discussed, such as bodily pain and its connection to mental health concerns. The manuscript would be strengthened by explicitly addressing the potential risks involved and detailing the measures taken to mitigate these risks during data collection, such as ethical safeguards, referral mechanisms, or support provisions.

Finally, the study currently lacks clearly articulated limitations and recommendations. It is recommended that these sections be added after the Discussion to acknowledge the study’s constraints and to offer directions for future research and practice.

Reviewer #3: Thank you, the authors, for researching a very important topic of “The lived and living experiences of having chronic pain and mental illness among Canadian veterans.”

The following are my comments that I feel you could elaborate more on.

Abstract:

The abstract is well written.

Introduction:

The introduction is also well elaborated, clearly showing the interplay between chronic pain and mental illness

Methods:

Elaborate more on how sampling procedure of the study participants (was it purposive or was it convenience sampling)

Clarify how the sample size of 20 participants was determined. (What guided you to use 20 study participants and not any other number?)

Sample and Recruitment:

First, state the inclusion criteria for the broader narrative inquiry study focusing on identity in Canadian Veterans with chronic pain. Then follow this with what you considered to recruit participants in this current study.

Data collection:

Asking veterans about the interplay between their mental illness and chronic pain might evoke painful memories during the interviews. How were such situations handled during your interviews?

Findings:

Write the abbreviation VAC in full (for a reader to understand better)

In the Demographic table of the participants, Migraines & Conflict morality are listed as examples of Mental Illness Diagnoses. Could you clarify more on these two diagnoses

Discussion:

The findings are well discussed, comparing and critiquing the existing literature.

**Do you want your identity to be public for this peer review?** For information about this choice, including consent withdrawal, please see our Privacy Policy .

Reviewer #1: No

Reviewer #2: No

Reviewer #3: No

---

## [Decision Letter · Decision Letter 1]

16 Mar 2026

The Lived and Living Experiences of Having Chronic Pain and Mental Illness Among Canadian Veterans: A Qualitative Descriptive Study

PMEN-D-25-00428R1

**Dear Dr. Majid,**

We are pleased to inform you that your manuscript 'The Lived and Living Experiences of Having Chronic Pain and Mental Illness Among Canadian Veterans: A Qualitative Descriptive Study' has been provisionally accepted for publication in PLOS Mental Health.

Best regards,

Kizito Omona, PhD

Academic Editor

PLOS Mental Health

Thank you for addressing all the concerns raised

Reviewer Comments (if any, and for reference):

Reviewer's Responses to Questions

**Comments to the Author**

Reviewer #1: All comments have been addressed

Reviewer #3: All comments have been addressed

Reviewer #4: All comments have been addressed

publication criteria ? Is the manuscript technically sound, and do the data support the conclusions? The manuscript must describe methodologically and ethically rigorous research with conclusions that are appropriately drawn based on the data presented.

Reviewer #1: Yes

Reviewer #3: Yes

Reviewer #4: Yes

3. Has the statistical analysis been performed appropriately and rigorously?

Reviewer #1: I don't know

Reviewer #3: N/A

Reviewer #4: N/A

4. Have the authors made all data underlying the findings in their manuscript fully available (please refer to the Data Availability Statement at the start of the manuscript PDF file)?

Reviewer #1: Yes

Reviewer #3: Yes

Reviewer #4: Yes

5. Is the manuscript presented in an intelligible fashion and written in standard English?

Reviewer #1: Yes

Reviewer #3: Yes

Reviewer #4: Yes

**Reviewer #1:** This manuscript presents a timely and meaningful qualitative exploration of how Canadian Veterans understand and experience the intersection of chronic pain and mental illness. The description of analysis could be further explained for better rigorous analytic approach.This manuscript presents a timely and meaningful qualitative exploration of how Canadian Veterans understand and experience the intersection of chronic pain and mental illness. The description of analysis could be further explained for better rigorous analytic approach.

**Reviewer #3:**  All the comments raised previously have been addressed by the authors. I have no other comments. All the comments raised previously have been addressed by the authors. I have no other comments.

**Reviewer #4:** I have read the original submission, the revised submission and the comments of all 3 reviewers. I applaud the authors for critically and thoroughly attending to the critiques. Overall, i find your revised manuscript a solid contribution to the existing literature. I have no further comments of offer.I have read the original submission, the revised submission and the comments of all 3 reviewers. I applaud the authors for critically and thoroughly attending to the critiques. Overall, i find your revised manuscript a solid contribution to the existing literature. I have no further comments of offer.

**Do you want your identity to be public for this peer review?** For information about this choice, including consent withdrawal, please see our Privacy Policy .

Reviewer #1: No

Reviewer #3: **Yes:** Philip AmanyirePhilip Amanyire

Reviewer #4: No
